# PateGail++: Utility Optimized Private Trajectory Generation with Imitation Learning

**Yingjie Ma, Bijal Bharadva, Xin Zhang & Joann Qiongna Chen**
Department of Computer Science
San Diego State University
San Diego, CA 92182, USA
{yma6999, bbharadva1752, xzhang19, jchen27}@sdsu.edu

## Abstract

Human mobility trajectory data supports a wide range of applications, including urban planning, intelligent transportation systems, and public safety monitoring. However, large-scale, high-quality mobility datasets are difficult to obtain due to privacy concerns. Raw trajectory data may reveal sensitive user information, such as home addresses, routines, or social relationships, making it crucial to develop privacy-preserving alternatives. Recent advances in deep generative modeling have enabled synthetic trajectory generation, but existing methods either lack formal privacy guarantees or suffer from reduced utility and scalability. Differential Privacy (DP) has emerged as a rigorous framework for data protection, and recent efforts such as PATE-GAN and PateGail integrate DP with generative adversarial learning. While promising, these methods struggle to generalize across diverse trajectory patterns and often incur significant utility degradation. In this work, we propose a new framework that builds on PateGail++ by introducing a *sensitivity-aware noise injection module* that dynamically adjusts privacy noise based on sample-level sensitivity. This design significantly improves trajectory fidelity, downstream task performance, and scalability under strong privacy guarantees. We further adapt our framework to the local differential privacy (LDP) setting, allowing individual-level protection without reliance on a trusted server. We evaluate our method on a real-world mobility dataset and demonstrate its superiority over state-of-the-art baselines in terms of privacy-utility trade-off.

## 1 Introduction

Human mobility data supports a wide range of critical applications, including urban planning (Ruan et al., 2020), intelligent transportation systems (Ma et al., 2013), human mobility analysis (Liang et al., 2021; Wang et al., 2021), and public safety monitoring (Gao et al., 2017). However, obtaining high-quality, large-scale mobility datasets remains challenging due to serious privacy concerns. Raw trajectory records can expose sensitive personal information, such as home addresses, daily routines, or social relationships, that may be exploited for malicious purposes if not properly protected.

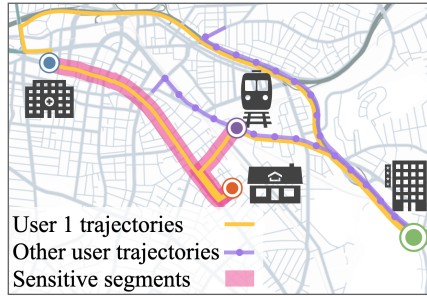

Figure 1: Trajectory segments that strongly reflect user-specific behaviors pose higher privacy risks than those that resemble common or overlapping ones.

Recent advances (Wang et al., 2023b;a; Roman et al., 2025; Zhang et al., 2025; Ma et al., 2024; Fan et al., 2024; Bouabba et al., 2024; Netzler & Lienkamp, 2024; Narayanan et al., 2024) in deep generative modeling have motivated the use of synthetic trajectory generation as a privacy-preserving alternative. While promising, many existing methods either lack formal privacy guarantees (Kim & Jang, 2022; Rao et al., 2020) or incur significant utility degradation and computational overhead (Du et al., 2023; Rao et al., 2020), limiting their practical applicability.

Differential Privacy (DP) has emerged as the gold standard for provable privacy protection (Dwork et al., 2006), and has been increasingly applied in data publishing (Asghar et al., 2020) and synthetic data generation (Jordon et al., 2018). One notable effort is PATE-GAN (Jordon et al., 2018), which combines generative adversarial networks with the Private Aggregation of Teacher Ensembles (PATE) (Papernot et al., 2017) framework to generate differentially private synthetic data. While promising, its utility remains limited, especially in complex spatial-temporal domains like mobility trajectories. On the other hand, PATEGAIL (Wang et al., 2023b) adapts DP mechanisms to the imitation learning setting, leveraging Generative Adversarial Imitation Learning (GAIL) (Ho & Ermon, 2016) while incorporating differential privacy into policy training. While PATEGAIL provides a promising direction for behavior modeling under DP, it suffers from reduced performance and challenges in generalizing to diverse trajectory patterns in real-world settings.

**Limitation of State-of-the-Art.** A critical limitation of current DP-based trajectory generation methods lies in their uniform treatment of all data points when applying privacy noise. These approaches typically inject the same amount of noise across all samples, regardless of their semantic or statistical sensitivity. However, not all trajectories pose the same privacy risk—some may contain highly identifying patterns, while others are inherently less sensitive. As shown in Fig. 1, trajectory segments that are behaviorally unique are more sensitive than those that overlap with many users' trajectories. Applying identical noise levels can therefore lead to unnecessary utility loss for low-risk samples and insufficient protection for highly sensitive ones. This rigid noise assignment limits scalability and weakens the overall privacy-utility trade-off.

**Our Approach.** We propose PATEGAIL++, a new differentially private imitation learning framework for trajectory generation. Our design introduces a sensitivity-aware noise injection module that dynamically adjusts noise levels according to the sensitivity of trajectory samples, offering stronger protection for high-risk segments while reducing unnecessary distortion in low-risk cases. We further extend this framework to the local differential privacy (LDP) setting, ensuring user-level protection without reliance on a trusted server. To stabilize training under adaptive noise, PATEGAIL++also incorporates Wasserstein GAN with Gradient Penalty (WGAN-GP) (Gulrajani et al., 2017a), which mitigates instability in discrete trajectory learning and accelerates policy convergence. *Our key contributions are as follows:*

- We design a differentially private generative model tailored for mobility trajectory data that better balances privacy and utility.
- We introduce a novel framework PATEGAIL++with a customizable sensitivity module that enhances trajectory realism while maintaining strict privacy guarantee.
- We extend our framework to the local differential privacy setting, allowing individual privacy protection without reliance on a trusted server.
- We conduct extensive experiments on a real-world dataset, demonstrating superior performance over existing baselines in privacy and utility trade-off.

## 2 PRELIMINARIES

### 2.1 HUMAN MOBILITY DECISION-MAKING

We consider a scenario with multiple users, each equipped with a personal mobile device that locally stores their historical mobility trajectories. These trajectories consist of sequential decision-making records that capture human movement behaviors, such as daily commuting or shopping. Formally, for each user $u \in \mathcal{U}$, the trajectory is represented as $T_u = \{(l_1, t_1), \cdots, (l_N, t_N)\}$, where $l_i$ and $t_i$ denote the location and timestamp at step $i$, respectively. These sequences are the result of the user's latent decision-making strategy (*i.e.*, policy) under spatial-temporal constraints.

Each *trajectory* describes a user's traversal through a sequence of spatial-temporal *states*, guided by a corresponding sequence of decisions (*i.e.*, *actions*). We define the *state* $s_t$ as a user's mobility history up to time $t$, *i.e.*, $s_t = \{(l_\tau, t_\tau)\}_{\tau \leq t}$, and the *action* $a_t$ action space is defined based on the widely-adopted exploration and preferential return (EPR) model (Song et al., 2010; Jiang et al., 2016). Core to this modeling framework are two functions, *i.e.*, *policy* and *reward*, that govern decision-making and need to be learned from trajectory data:

**Definition 1: Policy function** $\pi(a|s)$ defines the probability of choosing action $a$ given the current state $s$. This function governs how a user makes decisions at different situations.

**Definition 2: Reward function** $R(s, a)$ quantifies the implicit benefit or desirability of taking action $a$ at state $s$. This function explains the preferences that guide the user's behavior.

Human users are adapting policies for higher total reward in decision-making. Both the policy and reward functions are unobserved and need to be inferred from the user trajectories.

## 2.2 HUMAN STRATEGY LEARNING VIA IMITATION LEARNING

Generative Adversarial Imitation Learning (GAIL) (Ho & Ermon, 2016) is an imitation learning method that jointly learns the reward function and policy from expert demonstrations. It frames imitation as a minimax optimization problem, *i.e.*,

$$\min_{\pi_\theta} \max_{D_\phi} \mathbb{E}_{(s,a)\sim\pi_E}[\log D_\phi(s,a)] + \mathbb{E}_{(s,a)\sim\pi_\theta}[\log(1 - D_\phi(s,a))], \tag{1}$$

where $\pi_E$ denotes the expert (real) policy, and $D_\phi(s, a)$ is a discriminator trained to distinguish expert from generated behavior. The learned policy $\pi_\theta$ is trained using reinforcement learning to maximize expected rewards derived from the discriminator's output. The learned policy $\pi_\theta$ is optimized to fool the discriminator, effectively mimicking expert trajectories.

## 2.3 DIFFERENTIAL PRIVACY IN DECISION-MAKING

Differential Privacy (DP) (Dwork et al., 2006) is a formal framework for quantifying privacy guarantees. Let $\mathcal{D}$ and $\mathcal{D}'$ be two neighboring datasets that differ in a single record. A randomized algorithm $\mathcal{M}$ satisfies $(\varepsilon, \delta)$-differential privacy if, for all such neighboring datasets $\mathcal{D}, \mathcal{D}'$ and for all measurable subsets $\mathcal{E}$ of the output space,

$$\mathbb{P}[\mathcal{M}(\mathcal{D}) \in \mathcal{E}] \leq e^\varepsilon \cdot \mathbb{P}[\mathcal{M}(\mathcal{D}') \in \mathcal{E}] + \delta. \tag{2}$$

To ensure DP, noise is typically added using the Laplace mechanism:

$$\text{Lap}(x \mid \lambda) = \frac{1}{2\lambda} \exp\left(-\frac{|x|}{\lambda}\right), \tag{3}$$

where $\lambda$ is calibrated based on the global sensitivity and the privacy budget $\varepsilon$. The privacy budget controls how much information about a user's data can leak; smaller $\varepsilon$ means stronger privacy. Beyond noise addition, several fundamental properties of DP are critical when applying it to learning algorithms. Specifically:

**Sequential Composition.** The sequential composition states that combining multiple subroutines that satisfy DP for $(\varepsilon_1, \delta_1), (\varepsilon_2, \delta_2), \cdots$ results in a mechanism that satisfies $(\varepsilon, \delta)$-DP for $\varepsilon = \sum \varepsilon_i$ and $\delta = \sum \delta_i$.

**Post-processing.** Given an $(\varepsilon, \delta)$-DP algorithm $\mathcal{M}$, releasing $g(\mathcal{M}(\mathcal{D}))$ for any $g$ still satisfies $(\varepsilon, \delta)$-DP. That is, post-processing an output of a differentially private algorithm does not incur any additional loss of privacy.

**Zero-Concentrated Differential Privacy.** While $(\varepsilon, \delta)$-DP is the standard definition, Zero-Concentrated Differential Privacy ($\rho$-zCDP) (Dwork & Rothblum, 2016; Azize & Basu, 2024) provides a more refined privacy accounting framework. Its tighter composition guarantees make it particularly well-suited for iterative algorithms.

## 2.4 PROBLEM DEFINITION

*Differentially Private Trajectory Generation Problem.* Given a set of users $\mathcal{U}$ with private mobility trajectories $\{T_u\}_{u \in \mathcal{U}}$, the goal is to learn a global trajectory generation policy $\pi(a|s)$ such that: *1).* The generated synthetic trajectories are *utility-preserving*, *i.e.*, they statistically resemble the real trajectories and support downstream applications such as prediction and recommendation. *2).* The learning algorithm satisfies $(\varepsilon, \delta)$-*differential privacy*, meaning the presence or absence of any individual user's trajectory does not significantly affect the output.

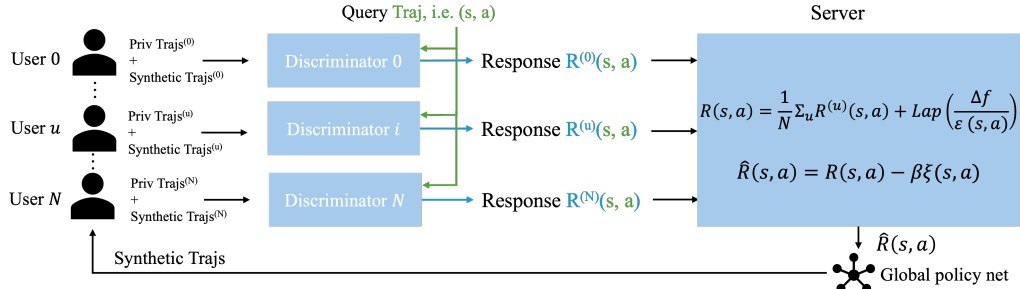

Figure 2: PATEGAIL++ framework.

*Challenges.* The proposed differentially private trajectory generation problem presents three key challenges: (**C1**) How can privacy budgets be allocated to reflect varying data sensitivity and ensure good model utility? (**C2**) How can we provide formal privacy guarantees under non-uniform noise, given that adaptive noise scaling complicates standard DP analysis? (**C3**) How can policy learning remain stable under adaptive noise injection in the federated learning setting?

*Federated Learning Setting.* In this work we adopt a federated data-access model following Wang et al. (2023b). Human mobility trajectories are sensitive, *e.g.*, GPS traces can reveal home and work locations, daily routines, and social patterns, and are often subject to legal or institutional restrictions that prohibit centralizing raw data. Federated learning thus provides an appropriate deployment framework: each user retains their trajectory data locally, and only differentially private reward signals are shared with the server. Our work focuses on improving the utility–privacy trade-off within this practically constrained federated setting.

## 3 METHOD

In this section, we address the differentially private trajectory generation problem by introducing the PATEGAIL++ framework as shown in Fig. 2. In this framework, each user device locally trains a discriminator on its private mobility data to evaluate the plausibility of synthetic trajectories, while the global policy is trained on the server using aggregated, differentially private reward signals derived from local discriminator outputs.

Three important components are proposed in PATEGAIL++: First, we introduce a sensitivity-aware module that identifies the privacy sensitivity levels of user trajectories in the training data, thereby addressing Challenge **C1** (see Sec.3.1). Second, we provide a formal theoretical guarantee for $(\varepsilon, \delta)$-differential privacy, thereby addressing Challenge **C2** (see Sec.3.2). Finally, to tackle Challenge **C3**, we design a reward function ensemble strategy that stabilizes policy learning under adaptively injected noise (see Sec.3.3).

### 3.1 SENSITIVITY-DRIVEN PRIVACY BUDGET ASSIGNMENT

In this framework, each user $u \in \mathcal{U}$ maintains a local discriminator $D_{\phi_u}$ trained to distinguish real from generated trajectories on-device. For each state-action pair $(s, a)$, the local reward estimate is $R^{(u)}(s, a) = D_{\phi_u}(s, a)$, and the server aggregates these signals via a Laplace mechanism,

$$R(s, a) = \frac{1}{|\mathcal{U}|} \sum_{u \in \mathcal{U}} D_{\phi_u}(s, a) + \text{Lap}(0, \lambda). \tag{4}$$

Here, the server sees all sample-wise rewards, and the $\text{Lap}(0, \lambda)$ ensures differential privacy for global policy updates.

To account for reward variability across users, PATEGAIL introduces a reward dynamics compensation term based on the variance of local rewards,

$$\hat{R}(s, a) = R(s, a) - \beta \cdot \xi(s, a), \text{ where } \quad \xi(s, a) = \sqrt{\text{Var}(D_{\phi_u}(s, a)) + \text{Lap}(0, \lambda_c)}. \tag{5}$$

This adjustment allows the global policy $\pi(a|s)$ to maximize a high-probability lower bound of the users' expected cumulative rewards, while satisfying $(\varepsilon, \delta)$-differential privacy.

Despite these contributions, PATEGAIL has important limitations. First, it applies a uniform privacy budget across all iterations, regardless of individual-wise as well as iteration-wise sensitivity. This neglects trajectory-specific privacy risks, potentially overprotecting benign data while underprotecting highly sensitive samples. Second, the added noise is not calibrated to local context or trajectory variability. As a result, the policy may suffer from degraded utility or lack robustness across heterogeneous mobility behaviors. These limitations motivate the need for a more fine-grained approach that is able to identify trajectory-level data sensitivity and adjust privacy budget accordingly.

**Trajectory Sensitivity Module.** To improve the privacy-utility trade-off in federated imitation learning, we introduce a trajectory sensitivity module that dynamically calibrates the amount of noise injected into the reward aggregation process based on the privacy sensitivity of samples each iteration. This mechanism is to enable fine-grained differential privacy guarantees. We also extend this to individual-level differential privacy guarantees later.

We estimate the privacy sensitivity of each state-action pair $(s, a)$ based on the output of local discriminators $D_{\phi_u}$. Intuitively, if a discriminator $D_{\phi_u}$ assigns a high reward $R^{(u)}(s, a) = D_{\phi_u}(s, a) \approx 1$, it indicates that the generated sample is highly indistinguishable from real user trajectories. Such samples are more likely to resemble unique or rare behaviors, making them more vulnerable to inference attacks. Therefore, they are considered more privacy-sensitive and require stronger protection.

To reflect this, we define the privacy sensitivity of a sample inversely with respect to its confidence margin from 1, *i.e.*,

$$\text{Sensitivity}(s, a) \propto \frac{1}{1 - \hat{R}(s, a) + \delta'},$$

where $\delta' > 0$ is a small constant to avoid division by zero. Based on this sensitivity score, we allocate a per-sample privacy budget $\varepsilon(s, a)$ such that:

$$\varepsilon(s, a) = \frac{\varepsilon \cdot w(s, a)}{\sum_{(s', a') \in \mathcal{D}} w(s', a')}, \text{where } w(s, a) = 1 - \hat{R}_p(s, a) + \delta'.$$

To avoid potential information leakage when adaptively allocating privacy budgets, the $\hat{R}_p(s, a)$ above is a differentially private pilot estimate of $\hat{R}(s, a)$. This ensures that samples with higher discriminator confidence (*i.e.*, more sensitive) receive a smaller share of the privacy budget, leading to stronger noise injection. The total privacy expenditure is constrained by:

$$\sum_{(s,a) \in \mathcal{D}} \varepsilon(s, a) = \varepsilon,$$

where $\mathcal{D}$ is the set of all state-action pairs queried during training.

This sensitivity-aware allocation enables stronger protection for high-risk samples while reducing unnecessary noise for low-risk ones, improving both privacy and utility under a fixed global budget.

**Interpretation of Sensitivity.** Our sensitivity module reflects how closely a generated state-action resembles behavioral patterns that are distinctive to an individual user. When a user's local discriminator $D_{\phi_u}$ assigns a high score, it indicates that the generated sample is highly similar to that user's real behavior. We therefore treat such samples as high-sensitivity and add stronger noise, preventing the server from learning or exploiting user-specific behavioral signatures. This design also preserves DP guarantees: incorporating external semantic cues (*e.g.*, location types or home-work labels) into the sensitivity calculation would depend on non-private information and violate the DP accounting assumption. By basing sensitivity solely on internal model signals, our formulation maintains end-to-end DP protection.

## 3.2 SENSITIVITY-AWARE REWARD AGGREGATION

For each state-action pair $(s, a)$, each user $u$ computes a local reward $R^{(u)}(s, a) = D_{\phi_u}(s, a)$. The server aggregates these using a sensitivity-aware Laplace mechanism:

$$R(s, a) = \frac{1}{N} \sum_{u=1}^{N} R^{(u)}(s, a) + \text{Lap}\left(\frac{\Delta f}{\varepsilon(s, a)}\right) \tag{6}$$

where $\frac{\Delta f}{\varepsilon(s,a)}$ equals to $\lambda$ in Eq. (4) if the total privacy are distributed evenly per sample $(s,a)$.

To account for reward variability across users, we adopt the dynamics compensation mechanism introduced in PATEGAIL.

$$\hat{R}(s,a) = R(s,a) - \beta \cdot \xi(s,a),$$

$$\text{where } \xi(s,a) = \sqrt{\text{Var}(R^{(u)}(s,a)) + \text{Lap}\left(0, \frac{\Delta f}{\varepsilon(s,a)}\right)} \tag{7}$$

**Policy Optimization.** The final perturbed reward $\hat{R}(s,a)$ is used to update the global policy $\pi(a \mid s)$ using any standard policy optimization method, such as Proximal Policy Optimization (PPO). The process repeats over multiple rounds of federated communication until convergence.

**Extending to LDP.** While our current framework enables fine-grained privacy control at the sample level with strong theoretical guarantees, it assumes a trustworthy server that can access raw rewards. To relax this assumption, we extend our framework to the local differential privacy (LDP) setting, where each user perturbs their own rewards before transmitting them to the server. In this way, the server never observes raw rewards from users (i.e., the outputs of individual discriminators). We further introduce a per-user privacy budget $\varepsilon^{(u)}(s,a)$, such that these budgets across users aggregate to the per-sample budget $\varepsilon(s,a)$. The aggregated reward is then computed as:

$$R(s,a) = \frac{1}{N} \sum_{u=1}^{N} \left( R^{(u)}(s,a) + \text{Lap}\left(\frac{\Delta f}{\varepsilon^{(u)}(s,a)}\right) \right) \tag{8}$$

To allocate per-user budgets, we define:

$$\varepsilon^{(u)}(s,a) = \frac{\varepsilon(s,a) \cdot w^{(u)}(s,a)}{\sum_{u=1}^{N} w^{(u)}(s,a)}, \text{where } w^{(u)}(s,a) \quad = 1 - \hat{R}^{(u)}(s,a) + \delta'.$$

Note that the weights can be calculated and shared using homomorphic encryption (Das, 2018) among users to protect individual privacy.

This design enables fine-grained privacy control at both the sample and individual-user levels, while preserving strong theoretical guarantees. As a result, our method improves utility in generated trajectories and downstream tasks, and broadens the applicability of the framework under more practical privacy assumptions.

## 3.3 WGAN-GP FOR STABLE POLICY LEARNING

The original PATEGAIL employs a cross-entropy–based discriminator, which often produces vanishing gradients and unstable updates when applied to discrete trajectory data. These issues are further amplified under DP-induced perturbations. To stabilize training, PATEGAIL++ adopts a Wasserstein GAN with Gradient Penalty (Gulrajani et al., 2017b), which minimizes the Wasserstein-1 distance between expert and generated distributions:

$$\min_{\pi_\theta} \max_{D_\phi} \mathbb{E}_{(s,a)\sim\pi_E}\left[D_\phi(s,a)\right] - \mathbb{E}_{(s,a)\sim\pi_\theta}\left[D_\phi(s,a)\right] - \lambda_{GP} \mathbb{E}_{\hat{x}}\left[(\|\nabla_{\hat{x}} D_\phi(\hat{x})\|_2 - 1)^2\right], \quad (9)$$

where $\hat{x}$ denotes samples interpolated between expert and generated state–action pairs, and $\lambda_{GP}$ controls the penalty strength.

Following the central reward design in Sec. 3.2, the critic output of each local discriminator is directly treated as the per-user reward, *i.e.*, $R^{(u)}(s,a) = D_\phi^{(u)}(s,a)$. These local rewards are then aggregated into $\hat{R}(s,a)$ through the sensitivity-aware Laplace mechanism. Compared to the log-based reward in GAIL, the Wasserstein critic provides smoother gradients and avoids reward saturation, leading to more stable aggregation and policy updates under adaptive noise injection.

We further apply spectral normalization as an optional regularizer to enforce Lipschitz continuity alongside the gradient penalty. This combination enhances the robustness of local discriminators, improves the fidelity of aggregated rewards, and ultimately accelerates policy convergence in PATEGAIL++.

## 3.4 PRIVACY RISKS ANALYSIS

Following PATEGAIL (Wang et al., 2023b), we measure the privacy risks in PATEGAIL++ using white-box membership inference attack (MIA), where the attack has full access to the target model. Specifically, let $\mathcal{T}_A$ denote the set of trajectories selected as attack targets. For each $\mathcal{T}_u \in \mathcal{T}_A$, the adversary extracts the reward corresponding to every state–action pair $(s, a)$ and constructs a feature vector $v_f$ from these values. This feature vector $v_f$ serves as input to a Random Forest classifier, whose objective is to determine whether $\mathcal{T}_u$ originates from the training set of the trajectory generation model. To train the classifier, an 70% of trajectories are drawn from the training set and 30% from outside the training set are used as positive and negative instances, respectively.

## 4 EVALUATION

**Dataset.** We evaluate PATEGAIL++ on two datasets: the processed Geolife dataset (Wang et al., 2023b) and the Telecom Shanghai dataset (Li et al., 2021; Guo et al., 2020; Wang et al., 2019). The Geolife dataset was previously used in PATEGAIL, and contains mobility trajectories of 83 users collected between April 2007 and October 2011. The Telecom Shanghai dataset contains more than 7.2 million records of accessing the Interent through 3,233 base stations from 9,481 mobile phones for six months. For both datasets, each record is obtained from GPS logs on users' mobile phones and includes latitude, longitude, and timestamp information.

**Implementation** We implemented PATEGAIL++ in `Python 3.10` using `PyTorch 2.1` with CUDA support, and conducted all experiments on an `NVIDIA A100 GPU` with 256 GB RAM. Training is based on Proximal Policy Optimization (PPO) (Schulman et al., 2017), with entropy regularization and gradient clipping for stability. Our system is fully configurable via YAML and supports automated logging, evaluation, and model checkpointing.

**Metrics.** To evaluate the quality of generated mobility trajectories, we adopt five statistical metrics at both trajectory and record levels, following (Wang et al., 2023b). At the trajectory level, *Radius* (of gyration) quantifies the dispersion of a trajectory around its center of mass, while *DailyLoc* measures the number of distinct locations visited. At the record level, *Distance* captures travel between consecutive trajectory points, *G-rank* reflects the normalized frequency of visits to globally popular locations across users, and *I-rank* measures user-specific visit frequencies to top locations averaged across individuals. Each metric is represented as a probability distribution over trajectories or records, and we employ the Jensen-Shannon Divergence (JSD) to quantify the discrepancy between real and synthetic distributions.

### 4.1 COMPARISON RESULTS WITH BASELINES

In this section, we evaluate PATEGAIL++'s utility performance using five key metrics: *Radius*, *DailyLoc*, *Distance*, *G-rank*, and *I-rank*. We assess robustness under varying noise levels. For clarity, all reported noise levels correspond to the uniform noise setting used in PATEGAIL; in contrast, PATEGAIL++ employs dynamic noise allocation, but we ensure that the total privacy budget is matched across methods. In addition to the privacy budget $\varepsilon$, we further evaluate privacy leakage through Membership Inference Attacks (MIA).

**Baselines vs. Federated/DP Setting.** The baselines (GAN (Goodfellow et al., 2020), SeqGAN (Yu et al., 2017), Time-Geo (Jiang et al., 2016), MoveSim (Feng et al., 2020), DiffTraj (Zhu et al., 2023)) are trained and evaluated in a centralized, non-federated, non-DP setting, whereas PATEGAIL and PATEGAIL++ operate in a federated setup with DP-style perturbations during reward aggregation. Therefore, the baseline numbers in Tab. 1 are reported to demonstrate *compatibility* of our method under stronger constraints, rather than to claim a strict apples-to-apples SOTA victory. See detailed description of the baseline methods in Appendix A.2 Despite these constraints, PATEGAIL++ attains competitive utility across metrics and strong semantic fidelity: for example, PATEGAIL++ achieves G-Rank = 0.0256 (better than MoveSim's 0.0387), with I-Rank comparable to MoveSim (0.0176 vs. 0.0173). In contrast, standard GAN suffers large errors (*e.g.*, DailyLoc = 0.5795, G-Rank = 1.0000). While SeqGAN and MoveSim yield smaller values on geometry-oriented metrics such as Radius/Distance, they do not consistently preserve ranking-based semantics; PATEGAIL++ better maintains high-level trajectory realism under federated/DP constraints.

Table 1: Comparison results with baselines. Approaches were implemented without noise on the training data, and the results of PATEGAIL and PATEGAIL++ are identical when the noise level is 0.

| Method | Radius | DailyLoc | Distance | G-rank | I-rank |
|---|---|---|---|---|---|
| PATEGAIL(++) | 0.0699 | 0.1046 | 0.0130 | **0.0256** | 0.0176 |
| GAN | 0.6931 | 0.5795 | 0.3191 | 1.0000 | 1.0000 |
| SeqGAN | 0.0757 | 0.0881 | 0.0115 | 0.0752 | 0.0329 |
| Time-Geo | 0.0544 | 0.4955 | 0.4116 | 0.1515 | 0.1461 |
| MoveSim | 0.0311 | **0.0293** | **0.0058** | 0.0387 | **0.0173** |
| DiffTraj | **0.0105** | 0.3792 | 0.0087 | 0.0501 | 0.0576 |

**Robustness Under Noise (matched protocol).** Tab. 2 and 3 demonstrate the robustness of PATE-GAIL++ under various noises. Under the federated training protocol, PATEGAIL++ demonstrates comparable performance to PATEGAIL at small noise levels (*e.g.*, Noise = 0.01), where the two methods achieve nearly identical utility across all metrics. At moderate noise (= 0.10), PATE-GAIL++ already shows advantages in higher-level semantics, reducing DailyLoc by $\approx 29\%$ (0.4914 vs. 0.6915) and lowering ranking errors (G-Rank 0.0278 vs. 0.0512, I-Rank 0.0607 vs. 0.2698), while maintaining similar Radius/Distance. At large noise (= 1.00), PATEGAIL++ clearly outperforms PATEGAIL, with lower DailyLoc and consistently better ranking fidelity. These results indicate that PATEGAIL++ is at least compatible with PATEGAIL at low-noise regimes and substantially more robust under stronger privacy perturbations.

Table 2: Comparison of PATEGAIL vs. PATEGAIL++ at various noise levels under Geolife dataset. PATEGAIL++ consistently outperforms PATEGAIL in DailyLoc, G-Rank and I-Rank.

| Metric | Noise = 0.01 | | Noise = 0.10 | | Noise = 1.00 | |
|---|---|---|---|---|---|---|
| | PATEGAIL | PATEGAIL++ | PATEGAIL | PATEGAIL++ | PATEGAIL | PATEGAIL++ |
| Radius | **0.0260** | 0.0409 | **0.0263** | 0.0392 | **0.0409** | **0.0409** |
| DailyLoc | 0.6837 | **0.6818** | 0.6915 | **0.4914** | 0.6070 | **0.5608** |
| Distance | **0.0045** | 0.0059 | **0.0049** | 0.0057 | 0.0059 | **0.0058** |
| G-Rank | 0.0512 | **0.0216** | 0.0512 | **0.0278** | 0.0216 | **0.0216** |
| I-Rank | 0.2358 | **0.1433** | 0.2698 | **0.0607** | 0.1158 | **0.1145** |

Table 3: Comparison of PATEGAIL and PATEGAIL++ at various noise levels under the Telecom Shanghai dataset.

| Metric | Noise = 0.01 | | Noise = 0.10 | | Noise = 1.00 | |
|---|---|---|---|---|---|---|
| | PATEGAIL | PATEGAIL++ | PATEGAIL | PATEGAIL++ | PATEGAIL | PATEGAIL++ |
| Radius | 0.3384 | **0.0071** | 0.3951 | **0.2752** | **0.4853** | 0.4978 |
| DailyLoc | **0.3412** | 0.4852 | 0.3402 | **0.3018** | **0.5554** | 0.6252 |
| Distance | 0.1251 | **0.0052** | 0.1789 | **0.0918** | **0.1520** | 0.2785 |
| G-Rank | 0.0668 | **0.0004** | **0.0668** | **0.0668** | 0.0668 | **0.0668** |
| I-Rank | **0.0199** | 0.2156 | 0.0325 | **0.0184** | **0.1163** | 0.1488 |

**Privacy Leakage.** We evaluate the vulnerability of PATEGAIL and PATEGAIL++ against both white-box and black-box membership inference attacks. For the white-box setting, we follow PATE-GAIL's setting and train a random forest classifier to distinguish members from non-members. For the black-box setting, we adopt LiRA (Carlini et al., 2022), which fits Gaussian loss distributions for each example and performs a likelihood-ratio test to infer membership. The white-box results for Geolife are presented in Tab. 4, and the corresponding LiRA results are shown in Tab. 5. Additional evaluations on the Telecom Shanghai dataset are provided in Appendix A.3.1. White-box MIA results show that PATEGAIL exhibits substantial privacy leakage at low noise (Accuracy = 0.6645, AUC = 0.7208 at Noise = 0.01), indicating that an adversary can reliably infer training membership. In contrast, PATEGAIL++ achieves near-random performance (Accuracy $\approx 0.51$, AUC $\approx 0.49$), effectively mitigating this risk. Importantly, this trend persists across different noise levels: while the attack on PATEGAIL remains above random until noise is very high, PATEGAIL++

consistently yields AUC values close to $0.5$, demonstrating strong resistance to inference attacks without sacrificing utility. LiRA reveals the same trend. As shown in Tab. 5, PATEGAIL++ consistently achieves lower attack accuracy (roughly a 10% reduction compared to PATEGAIL) while maintaining comparable AUC. This demonstrates that PATEGAIL++ offers stronger protection even under the state-of-the-art black-box threat model.

Table 4: Comparison of white-box MIA performance against PATEGAIL and PATEGAIL++ under different noise levels (70% members, 30% non-members) using Geolife dataset.

| Noise | PATEGAIL | | PATEGAIL++ | |
|---|---|---|---|---|
| | Accuracy | AUC | Accuracy | AUC |
| 0.01 | $0.6645 \pm 0.0104$ | $0.7208 \pm 0.0127$ | $\mathbf{0.5115} \pm 0.0098$ | $\mathbf{0.4962} \pm 0.0175$ |
| 0.10 | $0.6650 \pm 0.0239$ | $0.7273 \pm 0.0280$ | $\mathbf{0.4880} \pm 0.0168$ | $\mathbf{0.4846} \pm 0.0171$ |
| 1.00 | $0.5000 \pm 0.0175$ | $0.4972 \pm 0.0277$ | $\mathbf{0.4890} \pm 0.0103$ | $\mathbf{0.4921} \pm 0.0190$ |

Table 5: Comparison of LiRA performance against PATEGAIL and PATEGAIL++ under different noise levels using Geolife dataset.

| Noise | PATEGAIL | | PATEGAIL++ | |
|---|---|---|---|---|
| | Accuracy | AUC | Accuracy | AUC |
| 0.01 | $0.7000 \pm 0.1216$ | $\mathbf{0.6365} \pm 0.1722$ | $\mathbf{0.6088} \pm 0.1337$ | $0.6487 \pm 0.1788$ |
| 0.10 | $0.7000 \pm 0.1217$ | $0.5218 \pm 0.1800$ | $\mathbf{0.7000} \pm 0.1216$ | $\mathbf{0.5135} \pm 0.1598$ |
| 1.00 | $0.7000 \pm 0.1148$ | $\mathbf{0.6049} \pm 0.1786$ | $\mathbf{0.6088} \pm 0.1337$ | $0.6487 \pm 0.1788$ |

## 4.2 STUDY ON OTHER KEY HYPERPARAMETERS

We conduct an ablation study to evaluate the sensitivity of PATEGAIL++ to key hyperparameters and design choices. Specifically, we focus on the following gradient penalty coefficient introduced in Sec.3.3 and user subset ratio.

**Gradient Penalty Coefficient ($\lambda_{\mathbf{GP}}$).** To evaluate the effect of WGAN-GP regularization, we vary the gradient penalty coefficient across five settings: $\lambda_{GP} \in \{1, 5, 10, 15, 20\}$. As shown in Tab. 6, incorporating WGAN-GP consistently improves performance, with PATEGAIL++ outperforming PATEGAIL on nearly all metrics. For each noise level and metric, we also selected the best PATEGAIL++ configuration over $\lambda_{GP} \in \{1, 5, 10, 15, 20\}$ and compared it to PATEGAIL (See Fig.3 in Appendix A.3.2).

Table 6: Ablation study of different $\lambda_{GP}$ levels across noise levels.

| Noise | Metric | PATEGAIL | PATEGAIL++ | | | | |
|---|---|---|---|---|---|---|---|
| | | | w/o | $\lambda_{GP} = 5$ | $\lambda_{GP} = 10$ | $\lambda_{GP} = 15$ | $\lambda_{GP} = 20$ |
| 0.01 | Radius | 0.0260 | 0.0409 | 0.0409 | 0.0271 | 0.0409 | **0.0253** |
| | DailyLoc | 0.6837 | 0.6818 | **0.6018** | 0.6912 | 0.6789 | 0.6917 |
| | Distance | **0.0045** | 0.0059 | 0.0058 | 0.0049 | 0.0058 | 0.0048 |
| | G-Rank | 0.0512 | 0.0216 | **0.0216** | 0.0512 | **0.0216** | 0.0512 |
| | I-Rank | 0.2358 | 0.1433 | **0.1157** | 0.2694 | 0.1424 | 0.2700 |
| 0.10 | Radius | 0.0263 | 0.0392 | **0.0259** | 0.0268 | 0.0409 | 0.0264 |
| | DailyLoc | 0.6915 | **0.4914** | 0.6913 | 0.6917 | 0.6084 | 0.6912 |
| | Distance | 0.0049 | 0.0057 | **0.0048** | 0.0049 | 0.0059 | **0.0048** |
| | G-Rank | 0.0512 | 0.0278 | 0.0512 | 0.0512 | **0.0216** | 0.0512 |
| | I-Rank | 0.2698 | **0.0607** | 0.2704 | 0.2696 | 0.1158 | 0.2704 |
| 1.00 | Radius | 0.0409 | 0.0409 | 0.0271 | 0.0409 | 0.0258 | **0.0252** |
| | DailyLoc | 0.6070 | **0.5608** | 0.6652 | 0.6084 | 0.6916 | 0.6912 |
| | Distance | 0.0059 | 0.0058 | **0.0046** | 0.0059 | 0.0048 | 0.0048 |
| | G-Rank | 0.0216 | 0.0216 | 0.0512 | **0.0216** | 0.0512 | 0.0512 |
| | I-Rank | 0.1158 | **0.1145** | 0.2258 | 0.1158 | 0.2689 | 0.2686 |

**User Subset Ratio.** Since our approach trains one discriminator per user, we analyze the effect of varying the number of users involved during training. Specifically, we fix $\lambda_{GP} = 20$ and compare performance when using all, 80% and 40% of the available users. Due to space limit, we defer the results of user sebset ration in Appendix A.4.

### 4.3 EXTENDING TO LOCAL DIFFERENTIAL PRIVACY

In this section, we extend our framework to the local differential privacy (LDP) setting and evaluate its performance with and without sensitivity-aware aggregation at the individual level. We report results across the same five key metrics: *Radius*, *DailyLoc*, *Distance*, *G-rank*, and *I-rank*, as shown in Tab. 7. We denote the LDP variant of PATEGAIL++ with sensitivity-aware aggregation as PATEGAIL++$^+$ and the variant without sensitivity-aware aggregation as PATEGAIL++$^-$. Overall,

Table 7: Performance comparison between PATEGAIL++$^-$ and PATEGAIL++$^+$ under local differential privacy. Lower values indicate better performance.

| Noise | Metric | LDP | | Central | |
|---|---|---|---|---|---|
| | | PATEGAIL++$^-$ | PATEGAIL++$^+$ | PATEGAIL | PATEGAIL++ |
| 0.01 | Radius | 0.0409 | 0.0409 | 0.0260 | **0.0253** |
| | DailyLoc | 0.6789 | **0.6084** | 0.6837 | 0.6917 |
| | Distance | 0.0059 | 0.0059 | **0.0045** | 0.0048 |
| | G-Rank | **0.0216** | **0.0216** | 0.0512 | 0.0512 |
| | I-Rank | 0.1419 | **0.1158** | 0.2358 | 0.2700 |
| 0.10 | Radius | 0.0324 | **0.0262** | 0.0263 | 0.0264 |
| | DailyLoc | **0.6003** | 0.6916 | 0.6915 | 0.6912 |
| | Distance | **0.0045** | 0.0049 | 0.0049 | 0.0048 |
| | G-Rank | **0.0224** | 0.0512 | 0.0512 | 0.0512 |
| | I-Rank | **0.1241** | 0.2695 | 0.2698 | 0.2704 |
| 1.00 | Radius | 0.0409 | 0.0333 | 0.0409 | **0.0252** |
| | DailyLoc | 0.6084 | **0.3758** | 0.6070 | 0.6912 |
| | Distance | 0.0059 | **0.0046** | 0.0059 | 0.0048 |
| | G-Rank | **0.0216** | 0.0324 | 0.0216 | 0.0512 |
| | I-Rank | 0.1158 | **0.0726** | 0.1158 | 0.2686 |

PATEGAIL++$^+$ achieves comparable or superior performance to PATEGAIL++$^-$ across most settings. Moreover, PATEGAIL++ consistently pushes the privacy–utility frontier beyond PATEGAIL in both central and local settings.

## 5 CONCLUSION

In this work, we presented PATEGAIL++, a differentially private trajectory generation framework that adapts privacy budgets to trajectory sensitivity. By introducing sensitivity-aware allocation, we improve the privacy–utility trade-off over prior uniform-noise approaches. We further enhance training stability via WGAN-GP and extend the framework to the local differential privacy setting, enabling individual-level protection without assuming a trusted server. Experiments on real-world mobility data show that PATEGAIL++ consistently outperforms state-of-the-art baselines in both trajectory fidelity and robustness to membership inference attacks.

While PATEGAIL++ advances privacy-preserving trajectory generation, its sensitivity measure relies on discriminator confidence at the state–action level and may not fully capture semantic privacy risks, such as repeated visits to sensitive locations or long-horizon sequence patterns. Developing more nuanced sensitivity models, e.g., incorporating location-specific and long-term risk signals that treat places like hospitals as higher risk, is a promising direction for future work.

## 6 ACKNOWLEDGMENT

We thank the reviewers for the valuable suggestions. This project is supported by NSF IIS-2449864 and CNS-2451662.

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

# A    APPENDIX

## A.1    ETHICS AND REPRODUCIBILITY STATEMENTS

**Ethics Statement**   This research adheres to the ICLR Code of Ethics. All experiments are conducted on publicly available benchmark datasets and environments that do not involve human subjects or personally identifiable information. No new data collection was performed, and no sensitive or private information is included. The contributions of this work are methodological, aiming to advance machine learning. While reinforcement learning and related methods may be applied in safety-critical or socially sensitive domains, this paper does not directly address or deploy in such contexts. We have taken care to report results honestly, acknowledge limitations, and follow best practices for research integrity. No conflicts of interest or ethical concerns beyond standard research conduct arise from this work.

**Reproducibility Statement**   We have made every effort to ensure the reproducibility of our results. Detailed descriptions of the proposed algorithms, theoretical assumptions, and derivations are provided in the main text and appendices. Hyperparameter settings, model architectures, and training configurations are reported in full. Data preprocessing procedures and evaluation protocols are documented, and all datasets used are publicly available. Our source code and instructions for reproducing experiments are available at `https://github.com/yingjie1234/ PATEGAIL-PlusPlus-Utility-Optimized-Private-Trajectory-Generation`.

Together, these resources ensure that independent researchers can reproduce and verify our findings.

## A.2    BASELINE METHODS

We compare PATEGAIL++ against five baseline methods: GAN (Goodfellow et al., 2020), Seq-GAN (Yu et al., 2017), Time-Geo (Jiang et al., 2016), MoveSim (Feng et al., 2020), and Diff-Traj (Zhu et al., 2023). We describe each baseline in detail below.

- GAN (Goodfellow et al., 2020) directly generates full trajectories using a vanilla GAN architecture, training a generator and discriminator adversarially without explicit temporal modeling.
- SeqGAN (Yu et al., 2017) extends GANs to sequence generation by training the generator step-by-step via policy gradient, allowing it to produce trajectories one transition at a time.
- TimeGeo (Jiang et al., 2016) is a rule-based probabilistic mobility model grounded in the classical Exploration and Preferential Return (EPR) framework. It characterizes human movement using hand-crafted behavioral rules rather than learned representations.
- MoveSim (Feng et al., 2020) is a GAN-based trajectory generator that incorporates domain knowledge of human mobility regularities to enhance realism and statistical fidelity.
- DiffTraj (Zhu et al., 2023) is a diffusion-based trajectory generation model that learns continuous denoising dynamics over spatial-temporal embeddings. It achieves high-quality trajectory synthesis under a centralized, non-private training setting.

## A.3    MORE RESULTS

### A.3.1    PRIVACY LEAKAGE

As a complement to the Geolife results discussed in the main text, we further evaluate privacy leakage on the Telecom Shanghai dataset to confirm that the same trends persist across a different mobility distribution. The following tables report both white-box MIA performance and LiRA-based black-box performance under varying noise levels.

The Telecom Shanghai results show a similar overall pattern as in Geolife. Under white-box MIAs, PATEGAIL continues to leak membership information at moderate noise levels, reflected by elevated AUCs between 0.60 and 0.73, while PATEGAIL++ remains more stable and private in terms of AUCs. Moreover, PATEGAIL's attack performance fluctuates substantially across noise levels, whereas PATEGAIL++ exhibits more stable behavior with values consistently near chance. LiRA

leads to a similar conclusion: although both methods show some variance due to the dataset's complexity, PATEGAIL maintains higher attack accuracy and greater fluctuation across noise settings, while PATEGAIL++ remains reliably harder to exploit. Overall, the Telecom experiments reinforce that PATEGAIL++ provides stronger and more stable privacy protection than PATEGAIL under both white-box and black-box threat models.

Table 8: Comparison of white-box MIA performance against PATEGAIL and PATEGAIL++ under different noise levels (70% members, 30% non-members) using Telecom Shanghai dataset.

| Noise | PATEGAIL | | PATEGAIL++ | |
|---|---|---|---|---|
| | Accuracy | AUC | Accuracy | AUC |
| 0.01 | $\mathbf{0.5454} \pm 0.0730$ | $0.6263 \pm 0.0320$ | $0.6162 \pm 0.0621$ | $\mathbf{0.4881} \pm 0.0370$ |
| 0.10 | $0.6625 \pm 0.0801$ | $0.7299 \pm 0.0657$ | $\mathbf{0.6044} \pm 0.1095$ | $\mathbf{0.5380} \pm 0.1080$ |
| 1.00 | $\mathbf{0.5669} \pm 0.0641$ | $\mathbf{0.6022} \pm 0.0655$ | $0.5912 \pm 0.1256$ | $0.6149 \pm 0.1460$ |

Table 9: Comparison of LiRA performance against PATEGAIL and PATEGAIL++ under different noise levels using Telecom Shanghai dataset.

| Noise | PATEGAIL | | PATEGAIL++ | |
|---|---|---|---|---|
| | Accuracy | AUC | Accuracy | AUC |
| 0.01 | $0.5294 \pm 0.1020$ | $\mathbf{0.6409} \pm 0.1660$ | $\mathbf{0.4512} \pm 0.1220$ | $0.6808 \pm 0.1684$ |
| 0.10 | $\mathbf{0.5882} \pm 0.2210$ | $0.6858 \pm 0.1395$ | $0.6965 \pm 0.1320$ | $\mathbf{0.5840} \pm 0.2020$ |
| 1.00 | $0.7059 \pm 0.1690$ | $0.5452 \pm 0.2160$ | $\mathbf{0.6965} \pm 0.1320$ | $\mathbf{0.4962} \pm 0.2290$ |

### A.3.2 GRADIENT PENALTY COEFFICIENT

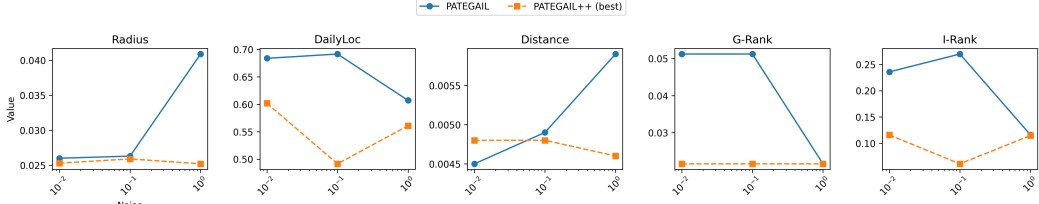

Figure 3: Comparison of PATEGAIL and the best-performing PATEGAIL++ configuration over $\lambda_{\text{GP}} \in \{0, 5, 10, 15, 20\}$, evaluated across various noise levels and metrics.

For each noise level and metric, we select the best PATEGAIL++ configuration over five different gradient penalty coefficients $\lambda_{\text{GP}} \in \{1, 5, 10, 15, 20\}$ and compare it to PATEGAIL. Fig.3 shows that PATEGAIL++ consistently outperforms PATEGAIL across all noise regimes and evaluation metrics. By adaptively allocating noise according to sample sensitivity and stabilizing training with WGAN-GP, PATEGAIL++ maintains high trajectory fidelity even under strong differential privacy constraints, whereas PATEGAIL deteriorates noticeably as noise increases. Across all metrics where lower is better (Radius, DailyLoc, Distance, G-Rank, and I-Rank), PATEGAIL++ either matches or surpasses PATEGAIL, often by a substantial margin in the moderate-noise setting where DP pressure is highest. Overall, PATEGAIL++ produces more stable, realistic, and consistent trajectories under differential privacy.

### A.4 USER SUBSET RATIO

The results in Tab.10 show that both PATEGAIL++ and PATEGAIL remain robust when user participation is reduced from 100% to 80%, showing only minor performance shifts. However, with a user participation rate of 40%, PATEGAIL++ generally yields lower differences in metric val-

Table 10: Performance comparison across different user percentages (40%, 80%, 100%) under varying noise levels.

| Noise | Metric | 40% | | 80% | | 100% | |
|---|---|---|---|---|---|---|---|
| | | PG | PATEGAIL++ | PG | PATEGAIL++ | PG | PATEGAIL++ |
| 0.01 | Radius | 0.0349 | 0.0409 | 0.0263 | 0.0252 | 0.0260 | 0.0253 |
| | DailyLoc | 0.3778 | 0.6084 | 0.6917 | 0.6916 | 0.6837 | 0.6917 |
| | Distance | 0.0046 | 0.0059 | 0.0049 | 0.0048 | 0.0045 | 0.0048 |
| | G-Rank | 0.0452 | 0.0216 | 0.0512 | 0.0512 | 0.0512 | 0.0512 |
| | I-Rank | 0.0589 | 0.1158 | 0.2703 | 0.2697 | 0.2358 | 0.2700 |
| 0.10 | Radius | 0.0409 | 0.0409 | 0.0265 | 0.0409 | 0.0263 | 0.0264 |
| | DailyLoc | 0.6818 | 0.6806 | 0.6795 | 0.6818 | 0.6915 | 0.6912 |
| | Distance | 0.0059 | 0.0059 | 0.0046 | 0.0059 | 0.0049 | 0.0048 |
| | G-Rank | 0.0216 | 0.0216 | 0.0512 | 0.0216 | 0.0512 | 0.0512 |
| | I-Rank | 0.1433 | 0.1431 | 0.2246 | 0.1433 | 0.2698 | 0.2704 |
| 1.00 | Radius | 0.0409 | 0.0281 | 0.0303 | 0.0392 | 0.0409 | 0.0252 |
| | DailyLoc | 0.6818 | 0.6919 | 0.6114 | 0.5364 | 0.6070 | 0.6912 |
| | Distance | 0.0059 | 0.0050 | 0.0046 | 0.0056 | 0.0059 | 0.0048 |
| | G-Rank | 0.0216 | 0.0512 | 0.0224 | 0.0092 | 0.0216 | 0.0512 |
| | I-Rank | 0.1433 | 0.2693 | 0.1384 | 0.0711 | 0.1158 | 0.2686 |

ues compared to PATEGAIL. This highlights PATEGAIL++'s stronger resilience under limited user availability.

