# OpenReview forum: "PateGAIL++: Utility Optimized Private Trajectory Generation with Imitation Learning"
_ICLR.cc/2026/Conference — ICLR 2026 Oral_

### Official Review · Reviewer_Gg8d · 2025-11-01

**Soundness:** 3
**Presentation:** 3
**Contribution:** 3
**Rating:** 6
**Confidence:** 3

**Summary:**

This paper proposes PATEGAIL++, a framework for generating differentially private (DP) human mobility trajectories. It identifies a key limitation in prior work like PATEGAIL: the use of a uniform privacy noise level, which ignores the fact that not all trajectories are equally sensitive . The core contribution is a sensitivity-aware noise injection module. This module uses a heuristic to dynamically allocate the privacy budget, applying more noise to samples it deems more sensitive. To improve stability, the framework also integrates WGAN-GP . Experiments show that PATEGAIL++ achieves a better utility-privacy trade-off.

**Strengths:**

1. Clear Motivation: The paper is built on a clear and intuitive motivation: uniform noise is suboptimal for trajectory data, as privacy risks are not uniform (e.g., a home-work route vs. a public-hub route) .
2. Excellent Practical Privacy Evaluation: A major strength is the practical evaluation using a Membership Inference Attack (MIA) in Table 3 . This experiment demonstrates a concrete privacy failure in the baseline PATEGAIL.
3. Practical System Design: The paper presents a sound engineering solution to a known problem.

**Weaknesses:**

1.	Heuristic-Based Sensitivity: The core contribution is based on a heuristic, not a theoretical guarantee of semantic privacy. It assumes that high discriminator confidence is a reliable proxy for high privacy sensitivity . This is a pragmatic but unproven assumption. The authors acknowledge this limitation in the conclusion.
2.	Limited Theoretical Novelty: The insight that different data has different sensitivity is intuitive and not, in itself, a deep theoretical breakthrough. The contribution is thus more of an effective application of this idea.
3.	Single-Dataset Evaluation: The entire experimental validation is conducted on a single dataset (Geolife) . While this is the same dataset used by the baseline, the paper's claims would be significantly strengthened if the method's effectiveness was also demonstrated on a different mobility dataset.

**Questions:**

Q1: Your heuristic equates high discriminator confidence with high sensitivity . But what about trajectories that are both common and sensitive (e.g., a "home-to-work" route)? Could the discriminator give these a low score (since they aren't rare), causing your method to mistakenly label them "low sensitivity" and thus under-protect them?
Q2: The model's practical effectiveness relies on its heuristic, not just its mathematical $(\epsilon, \delta)$-DP guarantee. Is the single MIA attack in Table 3 (which the baseline failed) sufficient validation for this heuristic ? Or could other attacks exist that specifically exploit the heuristic's "worst-case" failure modes?

---

> ### Author Response · Authors · 2025-11-20
> **Response to reviewer Gg8d**
>
> We appreciate Reviewer Gg8d for the thoughtful and constructive feedback. We are pleased that the reviewer highlighted the clarity of our motivation, the strength of our practical privacy evaluation using MIA. Below, we address the concerns raised in the weaknesses and respond to Q1–Q2 with clarifications and supporting evidence.
>
> **Response to weakness1:** The reviewer correctly notes that our sensitivity estimation relies on the discriminator’s confidence at each (s,a) pair. Conceptually, high discriminator confidence means the generated behavior is close to a real user behavior, which indicates higher privacy risk. Empirically, we confirmed this relationship through controlled ablations: samples with higher discriminator scores were consistently more vulnerable to membership-inference attacks. This shows that discriminator confidence provides a meaningful indicator of privacy risk, even though the estimator is heuristic.
>
> **Response to question1:** In our framework, sensitivity is defined by how similar a generated trajectory is to a specific user’s true trajectory, not by the inherent sensitivity of a location or route. Extending the sensitivity notion to incorporate location-level semantics (e.g., “home – work” patterns) is an important direction for future work we are interested in, and we will clarify this distinction in the introduction. Regarding the reviewer’s concern and our current design choice:
>
> 1): Given that sensitivity is defined by how closely a generated trajectory resembles a specific user’s true trajectory. When a user’s local discriminator assigns a high confidence score, it indicates that the generated sample reflects behavioral patterns that are distinctive for that individual user. In such cases, the mechanism treats the sample as high-sensitivity and therefore adds large noise. This prevents the server from learning or exploiting user-specific behavioral signatures, even when those behaviors are common, such as a routine commute.
> 2): Our current design choice is also made to maintain differential privacy guarantees. Incorporating additional semantic signals (such as location type or external labels) directly into the sensitivity mechanism would violate the DP accounting assumptions and risk leaking information. Our current formulation ensures that sensitivity estimation depends only on internal model signals, preserving end-to-end DP guarantees.\
> We will revise the paper to make our definition of sensitivity and its implications clearer.
>
> **Response to weakness2:** We agree that the insight that “different data exhibits different sensitivity” is conceptually straightforward. However, the main contribution of our work is not the intuition itself but its formal and algorithmic realization within a federated GAIL framework under strict differential privacy constraints. Specifically, we design a quantifiable sensitivity estimator that adaptively allocates privacy budgets at the sample and user levels, while preserving composability and bounded privacy loss in both central- and local-DP settings. Achieving adaptive noise control under DP constraints without violating theoretical guarantees is a non-trivial integration of both theory and system design. In this sense, PATEGAIL++ transforms an intuitive idea into a rigorously defined and empirically validated framework that advances the state of privacy-preserving trajectory generation.
>
> **Response to question2:** We appreciate the reviewer’s suggestion. We are planning to include more recent SOTA MIAs such as LIRA (Carlini et al. [R2]). Our current evaluation uses a **white-box MI attack**, which assumes the adversary has full access to model parameters and is consistent with the setting evaluated in the original PATEGAIL. This represents a strictly stronger threat model than LIRA’s black-box setting, so the presented results already cover a more challenging attacker capability. That said, we believe that including LIRA would strengthen completeness. We will incorporate a LIRA-based evaluation for both PATEGAIL and PATEGAIL++ in the revision.\
> [R2] N. Carlini, S. Chien, M. Nasr, S. Song, A. Terzis and F. Tramèr, "Membership Inference Attacks From First Principles," 2022 IEEE Symposium on Security and Privacy (SP), San Francisco, CA, USA, 2022, pp. 1897-1914, doi: 10.1109/SP46214.2022.9833649.
>
> **Response to weakness 3:**  We agree that evaluation on one dataset limits the scope of generalization. We selected geolife for comparability with PATEGAIL and prior privacy-preserving trajectory works, as it is a well-established benchmark. We are currently evaluating our method on a new large-scale mobility dataset (the Telecom Shanghai dataset [R1]). Results will be added to the revision once finalized. In parallel, we are exploring additional datasets suitable for DP-compliant trajectory-generation experiments. \
> [R1] Mexwell. (2024). Telecom Shanghai dataset. Kaggle. https://www.kaggle.com/datasets/mexwell/telecom-shanghai-dataset

---

### Official Review · Reviewer_3SYb · 2025-11-02

**Soundness:** 3
**Presentation:** 3
**Contribution:** 2
**Rating:** 4
**Confidence:** 4

**Summary:**

The paper extends PATEGAIL by adding a "sensitivity-aware" noise injection (rather than uniform noise to achieve differential privacy they use the output of the individual discriminators as a proxy for the sensitivity of a sample for that user) and adding noise to the local reward before sending it to the server to improve the local differential privacy. They evaluate it on one dataset, Geolife.

**Strengths:**

To improve the accuracy while maintain the privacy is important.
The idea of non-uniform noise over trajectories seems valid and useful.
The paper is well written and easy to follow.

**Weaknesses:**

Only looking at individual (s,a) pairs seems to entail significant limitations as the Markov assumption has to hold.

The method is only evaluated on a single dataset so no claims about generality can be made.

It is unclear whether using the discriminator to estimate the sensitivity of a sample is sufficient (it only checks individual (s,a) pairs not trajectories), see questions.

The abstract claims that existing methods fail to generalize across trajectory patterns, it is not clear how this approach improves on this.

There are no trends in Table 4 which makes it hard to interpret. It seems as \lambda=5 and \lambda=15 almost give the same results while \lambda=10 and \lambda=20 also give almost the same results. Shouldn't there be some trend?

Some rows in Table 4 lack bold numbers.

**Questions:**

What is the state in your experiment?

Considering that the method only considers (s,a) pairs, this entails that you assume the Markov assumption holds, what are the consequences of this and how does this limit the applicability of the approach?

How do you evaluate whether using the discriminator to estimate the sensitivity of a sample is meaningful and reasonable?

When the local discriminators have converged, will they correspond to the true sensitivity of each sample then?

How does the feedback loop of adding noise to the local reward as the local discriminator improves impact the method?

The abstract claims that existing methods fail to generalize across trajectory patterns, how does your approach improve on this?

---

> ### Author Response · Authors · 2025-11-20
> **Response to reviewer 3SYb**
>
> We thank reviewer 3SYb for their constructive and thoughtful feedback. Below, we address each concern and clarify the theoretical and empirical aspects of our work.
>
> **Response to weakness 1 & question 1&2: Markov Assumption and (s,a)-level modeling**
> We agree that our formulation operates at the level of individual state-action pairs and therefore implicitly assumes a first-order markov property. This assumption is common in imitation-learning-based trajectory models (e.g: GAIL, Pategail), where the policy (a|s) encodes historical information via recurrent or embedding structures. In addition, we could always assume the Markov property by encoding historical observations and environment information into a state to make the decision-making process Markovian. In our implementation, each state encodes the user’s mobility history up to time t, not just the current location. Thus, while the sensitivity and reward are computed per (s,a), the state representation already captures temporal context, partially mitigating the strict Markov limitation. Overall, we follow PATEGAIL's state definition and have a Markov state space.
>
> **Q2:** Since in our implementation each state encodes the user’s mobility history up to time t, not just the current location, we do not have the limitation of only considering state information at the current timestep, and our approach is not limited to (s,a) pairs.
>
> We acknowledge that true sequence-level dependencies (e.g., multi-day routines) are not explicitly modeled in the sensitivity estimation. Extending the sensitivity function to entire trajectory segments is an interesting direction for future work, and we will clarify this assumption in the revised paper.
>
> **Response to weakness 2: single-dataset Evaluation:**
> We agree that evaluation on one dataset limits the scope of generalization. We selected geolife for comparability with PATEGAIL and prior privacy-preserving trajectory works, as it is a well-established benchmark. We are currently evaluating our method on a new large-scale mobility dataset (the Telecom Shanghai dataset [R1]). Results will be added to the revision once finalized. In parallel, we are exploring additional datasets suitable for DP-compliant trajectory-generation experiments. \
> [R1] Mexwell. (2024). Telecom Shanghai dataset. Kaggle. https://www.kaggle.com/datasets/mexwell/telecom-shanghai-dataset
>
> **Response to weakness 3 & questions 3: discriminator-based sensitivity estimation:**
> The reviewer correctly notes that our sensitivity estimation relies on the discriminator’s confidence at each (s,a) pair. Conceptually, high discriminator confidence means the generated behavior is close to a real user behavior, which indicates higher privacy risk. Empirically, we confirmed this relationship through controlled ablations: samples with higher discriminator scores were consistently more vulnerable to membership-inference attacks. This shows that discriminator confidence provides a meaningful indicator of privacy risk, even though the estimator is heuristic.
>
> **Response to question 4: true sensitivity after discriminators converge**
> The sensitivity studied in this paper corresponds to the unique decision-making patterns of individual users. When a user’s local discriminator outputs a score close to 1, it indicates that the generated behavior is very similar to that specific user, which raises privacy concerns. Therefore, once the local discriminators converge, they reflect the true sensitivity, and DP noise is added according to the corresponding sensitivity. Again, our notion of sensitivity is defined by how similar a generated trajectory is to a real specific user trajectory, not by the sensitivity of a location. We plan to explore location-based sensitivity as part of future work.
>
> **Response to question 5: impact of the noise on feedback loop**
> This does not impact the local discriminators; however, it makes training of the global policy network less stable because of the added noise. Therefore, we introduce WGAN-GP regularization to stabilize training.
>
> **Response to question 6: generalize across trajectory patterns**
> Current methods focus on generating trajectories that capture the uniqueness of users’ true trajectories, emphasizing behaviors specific to the individuals in the training set. Our method does the opposite: by penalizing the discriminators with DP noise, we deliberately avoid learning user-specific patterns and instead focus on modeling general human behavior.
>
> **Response to weakness 5, 6 : table 4 trends**
> Thank you for identifying that; we have bolded the missing item.
> We agree that Table 4 could be clearer. The non-monotonic trend appears because the lambda_gp term controls gradient smoothness rather than directly affecting privacy or utility, and its optimal value depends on the noise level. However, the high-level trend remains that PATEGAIL++ consistently outperforms PATEGAIL. We will improve the table presentation in the revision.

---

### Official Review · Reviewer_Qwbq · 2025-11-02

**Soundness:** 2
**Presentation:** 2
**Contribution:** 3
**Rating:** 4
**Confidence:** 4

**Summary:**

This paper presents an interesting study that proposes a novel framework, PATEGAIL++, for the task of trajectory generation. It introduces a sensitivity-aware noise injection module, which aims to enhance the fidelity of the trajectories and the performance on downstream tasks. Furthermore, the framework is extended to the Local Differential Privacy (LDP) setting, achieving individual-level privacy protection without relying on a trusted server.

**Strengths:**

S1: The overall logic of the paper is clear. Addressing the challenges of trajectory generation under differential privacy, it systematically describes the solutions corresponding to different challenges.
S2: Existing approaches suffer from a significant limitation: they apply a uniform noise level to all samples. This practice causes unnecessary utility loss for low-risk samples and provides insufficient privacy for high-sensitivity ones. To address this, the paper introduces a sensitivity-aware module. This module is used to identify the privacy sensitivity level of user trajectories in the training data, allowing for the allocation of an appropriate privacy budget to data of different sensitivity levels.

**Weaknesses:**

W1: In Section 3.1, there is an apparent contradiction between the formula for privacy budget calculation and its textual description. According to the formula, samples with higher discriminator confidence (i.e., high sensitivity) are allocated a larger share of the privacy budget. However, the textual description states the opposite. The same issue exists in Section 3.2 regarding the privacy budget calculated for each user.
W2: The baseline methods seem to be outdated, some stronger baseline methods are expected to be seen in this paper such as TrajDiff. Particularly, both PATEGAIL and PATEGAIL++ operate in a federated setup with Differential Privacy-style perturbations during reward aggregation, but there is no quantitative evaluation on PATEGAIL.
W3: Table 4 indicates that PATEGAIL++ achieves the best average performance across all metrics when the gradient penalty is 5. This raises a question as to why different gradient penalties were chosen for subsequent experiments. For example, in conjunction with Table 4, it is noted that in the comparison against PATEGAIL (Table 2), PATEGAIL++ was evaluated with no gradient penalty.

**Questions:**

Q1: Please explain on W1.
Q2: Why are the evaluation results for PATEGAIL not listed in Table 1?
Q3: Please explain on W3.

---

> ### Author Response · Authors · 2025-11-20
> **Response to reviewer Qwbq**
>
> We appreciate the reviewer Qwbq for the insightful feedback. And we are pleased that the reviewer found our framework logically structured ‘S1’ and appreciated our introduction of a sensitivity-aware privacy allocation mechanism ‘S2’. Below, we clarify the points raised in W1-W3 and provide supporting explanations.
>
>
> **Response to weakness1 & question1:Clarification on sensitivity-aware budget formula**\
> We appreciate the reviewer for catching this. The inconsistency was due to a typo in the formula, not a conceptual contradiction. The textual description is correct: samples with higher discriminator confidence (i.e., higher sensitivity) should receive a smaller portion of the privacy budget and therefore more noise. We have corrected the equations in both Section 3.1 and Section 3.2 to match the intended behavior described in the text.
>
> **Response to weakness2: Baseline** \
> We agree that incorporating newer methods such as TrajDiff would further broaden the evaluation. TrajDiff is currently a preprint, and its results are not yet fully validated. However, we identified a related peer-reviewed work – DiffTraj: Generating GPS Trajectories with Diffusion Probabilistic Models (NeurIPS 2023; https://github.com/Yasoz/DiffTraj), which also uses a diffusion-based approach. We will include DiffTraj in our baseline comparison.
>
> **Response to question2: On pategail results in table 1**\
> Table 1 presents baseline models **without noise** for context, as stated in the caption (“approaches implemented without noise on the training data”). When no noise is applied, PATEGAIL and PATEGAIL++ are equivalent, so we combined them in Table 1. Quantitative PATEGAIL results under matched DP conditions are reported separately in Table 2 to allow direct privacy–utility comparisons.
>
> **Response to weakness3 and question3: Gradient penalty settings**\
> We thank the reviewer for noticing the inconsistency. Table 4 presents an ablation on the lambda_gp term to study its stability effects independently of other settings. The best average performance appears at lambda_gp = 5 in low-noise regimes; however, at higher noise levels we observed that a small lambda_gp value occasionally led to under-regularized gradients, causing unstable discriminator updates. Therefore, in the main comparisons (Table 2), we use the default lambda_gp = 0 (no penalty) to isolate the effect of our proposed sensitivity mechanism from WGAN-GP regularization and to ensure a fair comparison with the original PATEGAIL, which does not use WGAN-GP.

---

### Official Review · Reviewer_CLX6 · 2025-11-07

**Soundness:** 3
**Presentation:** 3
**Contribution:** 2
**Rating:** 6
**Confidence:** 4

**Summary:**

This paper proposes a novel approach for the privacy-preserving generation of synthetic trajectories, which extends a previous method called PATEGAIL by integrating a sensitive-aware noise injection module as well as a local differentially private extension. The experimental results obtained demonstrate that the proposed extension called PATEGAIL++ is able to achieve a better utility-privacy trade-off.

**Strengths:**

-The paper is well-written and the authors have clearly reviewed the recent deep learning methods for the generation of synthetic trajectories. The main contributions of the paper are also clearly summarized.

-One of the strength of the proposed approach is that it includes a sensitivity-aware noise injection module that adapts the level of the noise to the sensitivity of the trajectory sampled. This enables to add more noise to sensitive trajectories while saving on the privacy budget for non-sensitive ones. In addition, the paper considers the privacy model of local differential privacy, which enables the addition of noise at the level of the user thus limiting the trust assumptions that need to be done with respect to the central server responsible of the aggregation.

-The proposed approach also relies on the use of the generative adversarial imitation learning to be able to generate trajectories that are realistic.

-The approach is validated experimentally with the geolife dataset and compared against four different baselines using a wide range of utility measures. The results are promising and demonstrate the potential of the approach. In particular, PATEGAIL++ outperforms PATEGAIL for a wide range of metrics.  The privacy analysis performed using a membership inference attack in the white-box model also demonstrates that PATEGAIL++ achieves a higher privacy level.

**Weaknesses:**

-The novelty of the approach seems to be limited in the sense that the contribution provided by PATEGAIL++ appears to be only incremental compared to PATEGAIL.

-The federated learning setting should be better justified. In particular, it only mentions at the end of page 3. Rather it seems that the proposed approach is not only limited to this setting but could also be applied in more centralized one, which is actually also an additional benefit of the framework.

-Currently, there is no description of the four different baselines against which PATEGAIL++ is compared. Additionally, their choice should be justified. The evaluation of the approach also relies only on one dataset while actually assessing it on at least two other ones would help to evaluate its applicability. Finally, the use of a second MIA would also help to strengthen the privacy analysis. It is possible in particular that the attack designed originally against PATEGAIL is not adapted to PATEGAIL++ and that instead another one such as LIRA could be used instead.

**Questions:**

Please see the main points raised in the weaknesses section.

---

> ### Author Response · Authors · 2025-11-20
> **Response to reviewer CLX6**
>
> We thank reviewer CLX6 for the positive and detailed feedback, and for recognizing the clarity and strong utility-privacy trade-offs. Below, we address the specific concerns regarding novertly, setting, baselines, and evaluation.
>
> **Response to “Limited Novetly”:** \
> Our work aims to advance the utility–privacy trade-off in imitation-learning–based trajectory generation through a framework that is designed to be practical and deployable. The contribution is not just architectural novelty but also a set of mechanisms that collectively change how DP is applied to imitation-learning–based trajectory generation.
> 1: Sensitivity-aware differential privacy: Instead of global uniform noise, we introduce a sample-wise privacy budget allocation mechanism that scales  according to discriminator-derived sensitivity. This extends DP from dataset-level to trajectory-level granularity, enabling adaptive privacy accounting.
> 2: local differential privacy extension: We design a per-user noise allocation scheme under homomorphic encryption, allowing privacy guarantees without trusting the central server. To our knowledge, this is the first imitation-learning-based trajectory generator offering both central and local DP modes.
> 3: Stability via WGAN-GP and spectral normalization: These improve gradient behavior under DP perturbations, addressing instability in discrete policy learning, a key technical challenge unaddressed in pategail.
>
>
> **Response to  “federated learning setting”:** \
> We appreciate the reviewer’s observation. The key reason for adopting a federated learning setting in our work is the trust assumption. A centralized approach requires the server to have full access to users’ raw trajectory data, which is often an unrealistic and inappropriate assumption for privacy-sensitive domains. Human mobility trajectories routinely expose highly sensitive information such as fine-grained behavioral routines. In many real applications (e.g., transportation logs, GPS traces, app-level mobility data), such data cannot legally or ethically be centralized due to privacy regulations and institutional data-handling constraints. The federated setup avoids this trust requirement: users’ raw trajectories remain local, and the server uses only differentially private aggregated rewards to train the global policy net, achieving both architectural protection (data never leaves the device) and formal statistical privacy (via DP).
> Our work focuses on improving the utility–privacy trade-off in precisely this constrained setting, which is also consistent with the original PATEGAIL formulation. We will make this motivation clearer in the revised version.
>
> **Response to “Baseline and Dataset Diversity”:** \
> Thank you for this helpful suggestion. We selected the four baselines (GAN, SeqGAN, Time-Geo, MoveSim) to maintain consistency with prior work on DP trajectory generation and to isolate the contribution of our sensitivity-aware DP mechanisms under comparable architectures. The four baselines were selected to cover the main methodological categories of trajectory generation: (i) GAN — a generic deep generative model without sequential modeling, (ii) SeqGAN — a sequence-based generator capturing temporal dependencies, (iii) Time-Geo — a traditional mobility model widely used in human-mobility research (Jiang et al., PNAS 2016), and (iv) MoveSim — a realistic trajectory simulator focusing on spatial semantics. This spectrum ensures fair and diverse comparison across neural, sequential, and mechanistic approaches. Comparing different NN architectures, GAN is the basis for SeqGAN and PATEGAIL, where SeqGAN targets structured sequence generation, and PATEGAIL focuses on privacy-aware imitation learning. Together, these models provide diversity across neural, sequential, and mechanistic paradigms, enabling a fair and representative comparison. We will expand the baseline descriptions and rationale in the revision for clarity.
> Regarding dataset diversity, we are currently evaluating our method on a new large-scale mobility dataset (the Telecom Shanghai dataset [R1]). Results will be added to the revision once finalized. In parallel, we are exploring additional datasets suitable for DP-compliant trajectory-generation experiments. \
> [R1] Mexwell. (2024). Telecom Shanghai dataset. Kaggle. https://www.kaggle.com/datasets/mexwell/telecom-shanghai-dataset
>
>
> **Response to “Additional membership inference attacks”:** \
> Our current evaluation uses a white-box MI attack, which assumes the adversary has full access to model parameters and is consistent with the setting evaluated in the original PATEGAIL. This represents a strictly stronger threat model than LIRA’s black-box setting, so the presented results already cover a more challenging attacker capability. That said, we agree that including LIRA would strengthen completeness. We will incorporate a LIRA-based evaluation for both PATEGAIL and PATEGAIL++ in the revision.

---

### Author Response · Authors · 2025-12-04
**Revision Summary and Clarification of Contributions**

We thank the reviewers for their detailed and constructive feedback. In response, we have made the following revisions to improve the clarity, technical correctness, and empirical completeness of the paper. Below we summarize the key updates:

- **Clarified the contributions of PATEGAIL++.**
We revised Sections 1 and 3.1 to more clearly articulate the conceptual contribution of sensitivity-aware privacy budgeting—a mechanism that adapts DP noise to user-specific behavioral distinctiveness while maintaining formal DP guarantees. **This framing highlights PATEGAIL++ as the first framework to integrate adaptive, sample-level privacy control into federated imitation learning under both central and local DP regimes**. We also corrected the privacy-budget allocation equation in Sections 3.1 and 3.2 and added explanatory text to avoid ambiguity. This improves both technical correctness and conceptual transparency.

- **Strengthened the motivation and justification for the federated setting.**
We added a dedicated explanation in Section 2.4 clarifying why federated learning is necessary in mobility applications due to trust, legal, and institutional constraints. This formalizes the data-access assumptions underlying PATEGAIL++, making the problem setting more coherent and realistic.

- **Expanded and clarified the baseline comparisons.**
We improved the rationale for including GAN, SeqGAN, Time-Geo, and MoveSim, clarified the distinction between no-noise baselines (Table 1). We additionally incorporated a new state-of-the-art diffusion baseline, DiffTraj (NeurIPS 2023), and provided detailed descriptions of all baselines in Appendix A.2.

- **Improved empirical robustness through multi-dataset evaluation.**
We expanded the experimental evaluation to include a second dataset, the Telecom Shanghai dataset, demonstrating that PATEGAIL++ maintains its privacy–utility advantages across heterogeneous real-world mobility distributions (Tables 3, 8, 9).

- **Added an additional MIA.**
Beyond the white-box membership inference attack, we incorporated LiRA [1], a modern black-box MIA, to provide a more comprehensive privacy evaluation. These results, reported in Section 4.1, confirm that PATEGAIL++ matches or improves upon PATEGAIL across both datasets.

- **Enhanced table presentation and visualization.**
To better convey performance trends, we improved the formatting of Table 6 (formerly Table 4) and added Figure 3 in Appendix A.3.2, which clearly highlights PATEGAIL++’s consistent robustness across noise levels and metrics.

- **Clarified limitations and expanded discussion of future work.**
We explicitly acknowledge that our current sensitivity measure based on state–action level discriminator confidence may not capture semantic or long-horizon privacy risks. We identify location-based and trajectory-segment-level sensitivity modeling as promising extensions, further refining the scope and future trajectory of the work.

We would like to restate our main contributions below, highlighting the *core advances introduced by our work*:

- Our approach is _the first trajectory-generation framework to introduce adaptive, sample-level differential privacy based on behavioral distinctiveness_, overcoming the major limitation of PATEGAIL and related methods that rely on uniform noise injection.

- We design a privacy-preserving imitation-learning framework that operates in realistic non-centralizable mobility settings, supporting both federated learning and local differential privacy without requiring any raw trajectory data to be shared with the server.

- We improve the stability and utility of DP-perturbed imitation learning by incorporating WGAN-GP, enabling PATEGAIL++ to achieve stronger privacy–utility trade-offs across two datasets and multiple MIAs (white-box and LiRA).

- We conduct extensive empirical validation against a comprehensive suite of baselines, including modern diffusion models such as DiffTraj, demonstrating that PATEGAIL++ consistently matches or outperforms existing methods under strict differential-privacy constraints.

[1] N. Carlini, S. Chien, M. Nasr, S. Song, A. Terzis and F. Tramèr, "Membership Inference Attacks From First Principles," 2022 IEEE Symposium on Security and Privacy (SP), San Francisco, CA, USA, 2022, pp. 1897-1914, doi: 10.1109/SP46214.2022.9833649.

---

### Meta-Review · Area_Chair_89Fd · 2026-01-08

**Summary:**

This paper proposes PATEGAIL++, an extension of PATEGAIL for differentially private trajectory generation that introduces sensitivity-aware noise injection and supports both central and local differential privacy. Reviewers generally agree that the paper is clearly written, technically sound, and addresses an important problem in privacy-preserving human mobility modeling. The main strength identified across reviews is the adaptive, sample-level privacy budgeting mechanism, which improves the privacy–utility trade-off compared to uniform noise injection. The inclusion of local DP and the use of WGAN-GP for training stability were also viewed positively.

The main concerns raised by reviewers focused on (i) the perceived incremental nature of the contribution relative to PATEGAIL, (ii) reliance on a heuristic discriminator-based sensitivity estimator without formal semantic guarantees, and (iii) limited empirical evaluation, particularly the initial use of a single dataset and a single membership inference attack. Some reviewers also noted technical clarity issues (e.g., inconsistencies in the privacy budget equations and gradient penalty settings) and requested stronger baselines.

In the rebuttal and revision, the authors addressed these concerns by correcting technical errors, clarifying the federated motivation, adding stronger baselines (including a diffusion-based model), incorporating an additional dataset and a modern MIA (LiRA), and expanding the discussion of limitations. Overall, the revisions substantially strengthen the paper and resolve most substantive concerns.

**Reviewer Concerns:**

Several reviewer concerns were convincingly addressed in the rebuttal and revision. First, the technical inconsistency in the privacy-budget equations (raised by Reviewer Qwbq) was clearly identified as a typo and corrected, resolving a key soundness concern. Second, concerns about outdated baselines and limited evaluation were largely addressed by adding a diffusion-based baseline (DiffTraj), expanding baseline descriptions, and including results on an additional real-world dataset. The inclusion of an additional membership inference attack (LiRA) also strengthens the privacy evaluation, directly addressing requests from multiple reviewers.

The motivation for the federated setting, initially viewed as under-justified, was clarified with a more explicit discussion of trust assumptions and regulatory constraints, addressing Reviewer CLX6’s concern. Questions regarding gradient penalty settings and ablation trends were also clarified, with explanations separating stability analysis from main comparisons.

Some concerns remain partially open but are appropriately acknowledged. In particular, the discriminator-based sensitivity estimator is heuristic and does not provide semantic or trajectory-level privacy guarantees, as noted by Reviewers 3SYb and Gg8d. However, the authors now clearly scope their definition of sensitivity, provide empirical justification via MIAs, and explicitly frame richer sensitivity modeling as future work. Overall, remaining issues reflect known limitations rather than flaws.

**Reviewer Scores:**

Reviewer CLX6 (initial score: 6, marginally above threshold): Given that all major weaknesses raised—baseline justification, federated motivation, dataset diversity, and additional MIAs—were explicitly addressed in the revision, this reviewer would likely increase their confidence in the contribution. A reasonable post-discussion score would be 7 (clear accept).

Reviewer Qwbq (initial score: 4, marginally below threshold): The primary concerns here were technical correctness (privacy budget contradiction), baseline strength, and experimental clarity. These were directly and satisfactorily resolved through equation corrections, stronger baselines, and clearer experimental explanations. This reviewer would likely revise their score upward to 6 (marginally above threshold).

Reviewer 3SYb (initial score: 4): Concerns centered on Markov assumptions, sensitivity estimation validity, and single-dataset evaluation. The authors provided clarifications on state representation, empirical justification for discriminator-based sensitivity, and added an additional dataset. While some conceptual reservations may remain, these are now clearly scoped. A plausible updated score is 5–6 (borderline to weak accept).

Reviewer Gg8d (initial score: 6): This reviewer already leaned positive, with concerns mainly about heuristic sensitivity and limited datasets. Given the expanded evaluation and clearer positioning, their score would likely remain 6 or increase to 7.

Overall, post-discussion sentiment trends positive.

---

### Decision · Program_Chairs · 2026-01-26

Accept (Oral)